# Deep Learning-Based Optimal Smart Shoes Sensor Selection for Energy Expenditure and Heart Rate Estimation

**DOI:** 10.3390/s21217058

**Published:** 2021-10-25

**Authors:** Heesang Eom, Jongryun Roh, Yuli Sun Hariyani, Suwhan Baek, Sukho Lee, Sayup Kim, Cheolsoo Park

**Affiliations:** 1Department of Computer Engineering, Kwangwoon University, Seoul 01897, Korea; 9200heesang@gmail.com (H.E.); yulisun@telkomuniversity.ac.id (Y.S.H.); zhsjzhsj@gmail.com (S.B.); 2Digital Transformation R&D Department, Korea Institute of Industrial Technology (KITECH), 143 Hanggaulro, Ansan 15588, Korea; ssaccn@kitech.re.kr; 3School of Applied Science, Telkom University, Bandung 40257, Indonesia; 4Department of Leisure Sports, College of Ecological Environment, Kyungpook National University, Sangju-si 37224, Korea; ehduq132@gmail.com

**Keywords:** smart shoe, energy expenditure, heart rate, channel wise attention, DenseNet, accelerometer, gyroscope, pressure sensor, deep learning

## Abstract

Wearable technologies are known to improve our quality of life. Among the various wearable devices, shoes are non-intrusive, lightweight, and can be used for outdoor activities. In this study, we estimated the energy consumption and heart rate in an environment (i.e., running on a treadmill) using smart shoes equipped with triaxial acceleration, triaxial gyroscope, and four-point pressure sensors. The proposed model uses the latest deep learning architecture which does not require any separate preprocessing. Moreover, it is possible to select the optimal sensor using a channel-wise attention mechanism to weigh the sensors depending on their contributions to the estimation of energy expenditure (EE) and heart rate (HR). The performance of the proposed model was evaluated using the root mean squared error (RMSE), mean absolute error (MAE), and coefficient of determination ($R^{2}$). Moreover, the RMSE was 1.05 ± 0.15, MAE 0.83 ± 0.12 and $R^{2}$ 0.922 ± 0.005 in EE estimation. On the other hand, and RMSE was 7.87 ± 1.12, MAE 6.21 ± 0.86, and $R^{2}$ 0.897 ± 0.017 in HR estimation. In both estimations, the most effective sensor was the z axis of the accelerometer and gyroscope sensors. Through these results, it is demonstrated that the proposed model could contribute to the improvement of the performance of both EE and HR estimations by effectively selecting the optimal sensors during the active movements of participants.

## 1. Introduction

Wearable technologies have been continuously developed to improve the quality of human life and facilitate mobility and connectivity among users due to the rapid development of the Internet of Things (IoT). Its global demand is increasing every year [1,2,3]. Recently, several wearable devices, including wrist bands, watches, glasses, and shoes, have started enabling the continuous monitoring of an individual’s health, wellness, and fitness [4]. In particular, the coronavirus disease (COVID-19) pandemic highlighted the importance of remote healthcare delivery, resulting in further expansion of the wearable technology market [3,5]. This is because wearable devices could continuously collect and analyze the movement and physiological data of a user and provide appropriate feedback in function of users’ exercise information and health status.

The shoe is a useful wearable device that is easy to use, unobtrusive, lightweight, and easily available when doing outdoor activities [6,7,8,9]. Previous studies on shoes include gait type classification [9,10,11], step count [8,12,13], and energy expenditure (EE) estimation [14]. Three types of sensors (i.e., pressure, accelerometer, and gyroscope sensors) were equipped in the shoes to realize these tasks. These relatively low-cost sensors could be mounted in an unconstrained and convenient manner and record the movement information of users to estimate their physical behaviors.

The EE estimation was associated with physical activity (PA) which could influence an individual’s health conditions [15]. The PA level, which can be quantitatively assessed, is highly correlated with the risk of developing cardiovascular diseases, diabetes, and obesity [16,17]. In addition, there are only a few studies conducted on EE estimation using shoes compared to those on gait type classification and step counting. In addition, the accelerometer is one of the most commonly used sensors in shoes and other various devices for estimating EE [18,19,20,21,22].

In a previous study, a regression model was designed to estimate personal characteristics such as age, gender, height, weight, and BMI using accelerometer sensor data [18,20]. On the other hand, Vathsangam et al. used an accelerometer and a gyroscope sensor together to estimate EE, showing the improvement of the EE estimation by utilizing both sensor data [23]. In addition, a pressure sensor can also provide significant information to estimate EE. In a study conducted by Ngueleu et al., they predicted the number of steps taken by users using pressure sensors that were equipped to their shoes [13]. The results show that there was a high correlation between the number of steps and EE conducted by Nielson et al. [19]. Moreover, the pressure sensor could also be used along with the accelerometer sensor to improve the EE estimation. In [22], EE was estimated using barometric pressure and triaxial accelerometer sensors in various states such as sitting, lying, and walking. Additionally, Sazonova et al. estimated EE using the data from the triaxial accelerometer and five pressure sensors which were measured whilst the participants performed various activities such as sitting, standing, walking, and cycling [14].

The World Health Organization (WHO) reported that more than 30% of fatalities worldwide are caused by cardiovascular diseases (CVDs) [24]. The heart rate variability (HRV) is known as an important risk index for CVDs [25]. Accordingly, in recent years, various types of wearable devices have been developed (e.g., a watch-type device mounting electrocardiogram (ECG) or photoplethysmogram (PPG) sensors) to conveniently measure heart rate (HR). However, in an exercise environment, ECG is inconvenient to measure and PPG is affected by severe noise due to the movement. Instead of measuring the direct cardiac response, Lee et al. estimated HR from the activity information measured using an accelerometer and gyroscope sensors attached to the chest [26,27].

In recent years, advanced deep learning algorithms have been developed with the help of increasing computing power and a sufficient big dataset. There have been studies on the application of the deep learning approach to the wearable technology [28,29,30], where the algorithm performed well in regression and classification problems using physiological sensor data [21,31,32]. Staudenmayer et al. reported that an artificial neural network (ANN) model can predict the EE information using the accelerometer signals [21]. However, they extracted hand-crafted features from the signals and fed them into the ANN model, which are challenging to extract and suboptimal in distinguishing sophisticated patterns in the signal due to its fixed model-based approach. Zhu et al. successfully improved the accuracy of the EE estimation using convolutional neural network (CNN) by extracting subtle patterns from the accelerometer and heart rate signals [33].

In the studies [23,33], the multichannel data from the accelerometer and gyroscope sensors were simultaneously analyzed to estimate EE and HR, which could have been improved by considering the significance of each channel data. It is important to investigate which channel’s data are the most significant when multivariate input data can be obtained from multichannel sensors to derive the target variable. In recent studies, a method to determine the weight for each input channel to a neural network was suggested using the channel-wise attention based on deep learning techniques [34,35,36].

This study investigated the novel approach in estimating EE and HR using wearable sensors. A smart shoes system was selected for the convenience of users rather than the direct cardiac response measurement system, owing to its unobtrusive and natural manner of measuring the activities of users in their daily life. Conventionally, smart shoes are equipped with three types of sensors (i.e., pressure, accelerometer, and gyroscope) to produce multichannel data. Moreover, a deep neural network model was designed to infer EE and HR information from the multichannel data without using model-based hand-crafted feature extraction methods, and the attention mechanism provides appropriate weights to the input channels of the networks to improve the inference performance. Additionally, the weights decided by the attention algorithm provide the importance of three different sensors and their channels to the estimation of the physiological variations, EE, and HR. This could also enhance our understanding of the designed deep neural network structure, also known as explainable artificial intelligence [37].

The rest of this study is organized as follows. Section 2 discusses the design and data collection process of the experiment. Section 3 introduces the structure and the learning process of the proposed deep learning model. In addition, Section 4 discusses the results of HR and EE estimations using the proposed model and statistical analysis of the attention weights of sensors used as inputs. The results presented in Section 4 are discussed in Section 5 using the existing related studies. Finally, this study is concluded in Section 6.

## 2. Materials and Methods

### 2.1. System Overview

Figure 1 shows the overall system architecture for EE and HR estimation. The participant in the study wore a calorimeter (K4b2, Cosmed, Italy) and a chest strap (H10, Polar, Finland) for EE and HR measurements. Moreover, for the signal detection of walking and running, four film-type pressure sensors on each foot and a sensor (BMI160, Bosch Corp, Reutlingen, Germany) capable of the simultaneous measurement of 3-axis accelerometers and gyroscopes were mounted between the shoe’s insole and outsole (Salted, Korea). Their locations are shown in Figure 2. In the figure, the locations of the pressure sensors are illustrated on the anatomical sketch. All sensor signals were simultaneously measured as the participant ran on the treadmill and predicted the EE and HR by using the deep learning model. The predictions were evaluated using the measurements from the calorimeter and chest strap.

### 2.2. Experiments

Ten healthy adult males (age: 22.5±1.8 years old, height: 172.9±3.5 cm, weight: 69.3±4.9 kg, foot size: 264±4.6 mm) without musculoskeletal and nervous system abnormalities were recruited for this study. Written informed consent was obtained from all participants. The study design and protocol was approved by the Institutional Review Board (IRB No. P01-201908-11-002).

The participants wore shoes equipped with pressure, accelerometer, and gyroscope sensors in their stable states before the experiment. In addition, as shown in Figure 3, they wore an HR strap and a calorimeter for measuring the HR and EE, respectively. Each participant ran on an electric treadmill at a speed varying from 3 to 10 kph, which increased by 1 kph per every 2 min (total 16 min ran) and they were instructed to run at a constant speed as much as possible. Each shoe data type of the participants (gyroscope, accelerometer, and pressure sensor data), HR, and EE were simultaneously recorded during the experiment. The shoes data were obtained using a smartphone app at a sampling rate of 33.3 Hz, while the HR and EE were acquired using the K4b2 software and recorded when the participant exhaled.

### 2.3. Data Preparation

Figure 4 shows the overall data preparation process for model training proposed in this study. It was difficult to determine the exact HR and EE that correspond to the data of the sensors attached on the shoes because the sampling periods of HR and EE recording (approximately 2–5 s) were not the same as those of the shoes’ sensors (30 ms). Therefore, HR and EE data were resampled to match the sampling period of the data of the sensors attached on the shoe using a linear interpolation method, as shown in Figure 5. In addition, the obtained data were standardized for the efficient learning of the proposed deep learning model and reduced adverse effects of outliers [38]. The input sample used by the proposed deep learning model consisted of 20 channel data (four points’ pressure on the left and right shoe each, triaxial accelerometer, and gyroscope) which were 10 s long, and the average values of HR and EE for 10 s were used as its label. The 10-s sample was overlapped to the next one by 1 s. The total number of samples was 9600. Moreover, Figure 6 shows the distributions of the HR and EE labels of 10 participants.

## 3. Proposed Model

Figure 7 shows the overall structure of the model proposed in this study. The channel-wise attention layer, which is described in Section 3.1, provides weights to the significant channels of the sensors mounted on the shoes to accurately estimate HR and EE. The weighted signals by the attention layer pass using DenseNet [39], which is a CNN-based model known to be excellent in extracting key features from input data and generating spatial feature vectors that are discussed in Section 3.2. The bidirectional gated recurrent unit (GRU) [40] models the temporal relationship among the feature vectors, enabling an intuitive and efficient learning by observing the variations of input data over time (described in Section 3.3). Furthermore, the global average pooling (GAP) [41] layer compresses the information of the spatiotemporal features vectors and output values of HR and EE (described in Section 3.4). The advantages of the proposed model are as follows:The manual feature extraction process is not necessary since a fully automated end-to-end deep learning model was applied;The spatiotemporal characteristics of the multivariate time-series data that is complex to process could be effectively extracted using DenseNet and bidirectional GRU (Bi-GRU);The importance of each channel in estimating HR and EE could be quantified using the channel-wise attention method, and it can explain the optimal sensors for the task.

**Figure 7 sensors-21-07058-f007:**
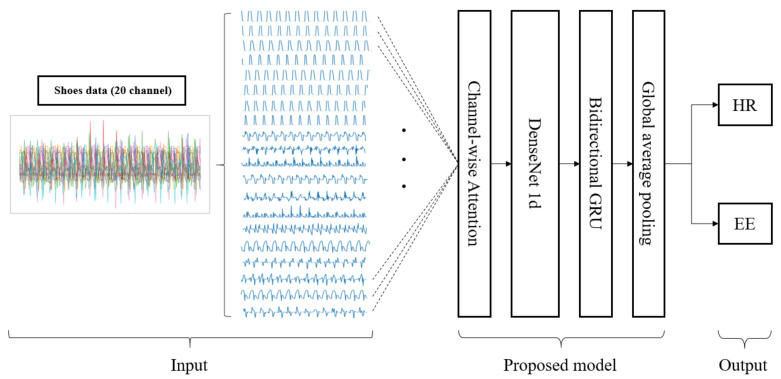
Structure of the proposed model. The shoe data from 20 channel sensors are fed into the input of the model and the channel-wise attention layer increases the intensity of the significant channels. The spatial features from the multi-channel data are extracted using DenseNet, and the temporal features are produced through Bi-GRU. Finally, HR and EE are estimated after the global average pooling (GAP) layer.

### 3.1. Channel-Wise Attention

It is difficult to extract the key features corresponding to HR and EE from the complex multivariate show data consisting of 20 channels. The conventional deep learning models train all input data with equal weights. This could deteriorate the learning efficiency of the model owing to the unnecessary and redundant information. However, the deep learning model could be efficiently trained by minimizing the unnecessary information in the input data and maximizing the significant information to the task. The attention mechanism is an optimized way of making this possible. In this study, we aimed to find and verify the optimal sensors for the estimation of HR and EE using the channel-wise attention expressed as follows:(1)$$O=\sigma \left(S_{in}W_{in}+b_{in}\right),$$
(2)$$S_{att}=Att⊗S_{in},$$
(3)$$Att=Average_{t}\left(O\right),$$
where *O* is calculated with the 20 channel signal $S_{in}=\left[s_{1},s_{2},\cdots s_{i}\right]\left(s\in R^{t},S_{in}\in R^{t\times i}\right)$, a trainable weight matrix $W_{in}$$\left(W_{in}\in R^{i\times i}\right)$, a bias $b_{in}$$\left(b_{in}\in R^{t}\right)$, and a non-linear activation function $\sigma \left(\cdot \right)$. In addition, *t* is the time length of a sample and *i* the number of channels. A sigmoid function [42] was chosen in this study for the activation function. $Att\left(Att\in R^{i}\right)$ represents the attention weights, which is calculated by the average of *O* across the time axis using the $Average_{t}\left(\cdot \right)$ function. Finally, the signal $S_{att}\left(S_{att}\in R^{t\times i}\right)$ is derived by multiplying Att and $S_{in}$ element-wise operation, which is expressed as ⊗.

### 3.2. DenseNet

DenseNet has yielded excellent performance in various image classification tasks [43,44,45,46]. Moreover, it avoids information dilution unlike other CNN-based models by concatenating the feature map output and input data in each convolutional layer. In addition, this method achieved higher performance with fewer parameters than that of the other models [39]. Therefore, DenseNet was used as a feature extractor in this study. The convolution layer was changed from two-dimensional (2D) to one-dimensional (1D), as shown in the Figure 8, since the shoe data are time-series data. In addition, the GAP layer was removed from its connection with the Bi-GRU layer in the last layer. The input to DenseNet $S_{att}$ is produced from the channel-wise attention layer. The output is represented as follows:(4)$$F_{dense}=DenseNet\left(S_{att}\right)$$

The final output vector is $F_{dense}=\left[x_{1},x_{2},\cdots ,x_{T}\right]\left(x\in R^{c_{1}},F_{dense}\in R^{c_{1}\times T}\right)$, where *T* is the time length compressed by the pooling layer and $c_{1}$ is the number of output of the last convolution layer, because the DenseNet used in this study has no GAP in the last layer.

### 3.3. Bidirectional Gated Recurrent Unit

In the proposed model, the temporal features are extracted from the output of DenseNet, $F_{dense}$, using the Bi-GRU layer defined in Equation (Equation 5). GRU is one of RNN models with powerful modeling capabilities for long-term dependencies. On the other hand, long short-term memory (LSTM) [47] is another popular RNN model. Between the two, GRU has a more efficient structure with fewer parameters [40]:(5)$$F_{bigru}=BiGRU\left(F_{dense}\right)$$

The hidden vector of Bi-GRU, $F_{bigru}=\left[h_{1},h_{2},\cdots ,h_{T}\right]\left(h\in R^{c_{2}},F_{bigru}\in R^{c_{2}\times T}\right)$, was obtained from $F_{dense}$, where $c_{2}$ is the size of the hidden unit of the GRU, as shown in Figure 9. Moreover, the internal structure of the GRU cell is shown in Figure 10. The operation is elaborated as follows:(6)$$r_{t}=\sigma \left(W_{r}*\left[h_{t-1},x_{t}\right]\right)$$
(7)$$z_{t}=\sigma \left(W_{z}*\left[h_{t-1},x_{t}\right]\right)$$
(8)$${\tilde{h}}_{t}=\tanh\left(W_{h}*\left[r_{t}*h_{t-1},x_{t}\right]\right)$$
(9)$$h_{t}=\left(1-z_{t}\right)*h_{t-1}+z_{t}*{\tilde{h}}_{t}$$

In Equations (Equation 6)–(Equation 9), $r_{t}$ and $z_{t}$ are the update gate and the reset gate vectors for an arbitrary time point t∈[1,T], respectively. The update gate determines how much information from the past and the present will be used to generate new information. The reset gate specifies which information to retain from the past information at the time t−1. Moreover, ${\tilde{h}}_{t}$ is a candidate state, which decides the amount of current information to be learned using the result of the reset gate. $W_{z}$, $W_{r}$, and $W_{h}$ are the trainable weight vectors of each gate. In addition, σ(·) and tanh(·) are the sigmoid and hyperbolic tangential functions, respectively. Furthermore, * denotes the element-wise multiplication.

Bi-GRU could simultaneously utilize both the past and future information, creating more useful features than unidirectional GRU. This is implemented as a forward and backward layer, as shown in Figure 9. The final output $h_{t}$ of Bi-GRU is determined by the concatenation of the two vectors when the forward and backward hidden vectors are represented as $\vec{h_{t}}$ and $\overset{\leftarrow }{h_{t}}$, respectively:(10)$$h_{t}=\vec{h_{t}}⊕\overset{\leftarrow }{h_{t}}$$

### 3.4. Global Average Pooling

In the proposed model, the GAP layer was designed in the last layer instead of the fully connected (FC) layer, which tends to overfit on the training data. This could degrade the generalization performance of the networks. On the other hand, no additional parameters were required since the GAP layer only calculates the average across the final output vectors of the network, reducing the overall network size and preventing overfitting. The final predicted target variables (i.e., HR and EE) using GAP are calculated as follows:(11)$$Target=\left(\frac{1}{T}\sum_{t=1}^{T}F_{bigru}\right)W_{out}+b_{out}.$$

### 3.5. Model Training Environment

The proposed model uses leave-one-subject-out (LOSO) cross-validation to evaluate the robustness and generalizability in an inter-subject analysis. The data of 9 subjects out of 10 subjects were used as the training set and the data of the remaining 1 subject were used as the testing set, which was repeated for all subjects. The mean and standard deviation of performance for each subject were calculated and described in Section 4. The Adam [48] optimization (learning rate = $10^{-3}$) was used to train the model, and the batch size was empirically set to 16. The initial weights of the networks were set at random and the loss function was designed based on the mean squared error (MSE). An early stopping method was applied to find the optimal model when there is no significant improvement in the validation loss of 20 epochs in a total of 150 training epochs. Furthermore, 4.2 GHz Intel Core i7 processor (Intel, Santa Clara, CA, USA) and NVIDIA GeForce RTX 2080Ti (NVIDIA corporation, Santa Clara, CA, USA) (with 11 GB VRAM), which are the computing environment for network training, were used. The model was implemented in Keras deep learning framework with TensorFlow backend.

## 4. Results

The results of the proposed model were evaluated in the following three aspects:Performance evaluation of the HR and EE estimation models;Performance analysis with and without the attention mechanism;Analysis of the channel significance using the attention weight;

The performance of the model was evaluated using several indicators. The root-mean-square error (RMSE), mean absolute error (MAE), and coefficient of determination ($R^{2}$) between the predicted and ground truths were calculated. Additionally, a Bland–Altman plot [49] was also presented. The formula of the evaluation indices are as follows:(12)$$RMSE=\sqrt{\frac{1}{N}\sum_{i=1}^{N}{\left(y_{i}-{\hat{y}}_{i}\right)}^{2}},$$
(13)$$MAE=\frac{1}{N}\sum_{i=1}^{N}\left|y_{i}-{\hat{y}}_{i}\right|,$$
(14)$$R^{2}=1-\frac{\sum_{i=1}^{N}{\left(y_{i}-{\hat{y}}_{i}\right)}^{2}}{\sum_{i=1}^{N}{\left(y_{i}-{\bar{y}}_{i}\right)}^{2}},$$

In Equations (Equation 12)–(Equation 14), *N* is the total number of test samples, $y_{i}$ is the ground truth, ${\hat{y}}_{i}$ is the predicted value, and ${\bar{y}}_{i}$ is the average value of $y_{i}$.

### 4.1. Energy Expenditure Estimation

#### 4.1.1. Proposed Model Performance

Table 1 shows the EE estimation performance using the proposed model. The pressure, accelerometer, and gyroscope sensor data were all used as input data. The RMSE between the predicted and ground truths was 1.05 ± 0.13, MAE was 0.83 ± 0.12, and $R^{2}$ was 0.922 ± 0.005. Figure 11 illustrates the predicted and ground truths across time for the best- and worst-case scenarios using the proposed model.

#### 4.1.2. Channel-Wise Attention Effectiveness

Analyzing what kind of sensors are helpful in estimating HR or EE using the channel-wise attention mechanism is the main objective of this study. This process could not be significant if the channel-wise attention degrades the performance of the model. The averaged results among the 10 participants are shown in Table 2 and Figure 12.

The proposed model using the channel-wise attention in EE estimation achieved higher performance in RMSE and MAE compared to that without the channel-wise attention. In addition, a significant improvement was confirmed using the one-tailed paired-sample *t*-tests (p<0.05). Although the $R^{2}$ of both models were quite close, the performance of the attention model is more stable with a standard deviation that is less than 0.005.

In Figure 12, the orange line is the limit of agreement (LOA), which means that the difference between the actual and estimated values is within the range of [lower LOA, upper LOA]. At the 95% confidence interval (±1.96SD), in this study, the LOA was [−1.56, 1.93] and [−2.42, 3] in the model with and without the channel-wise attention, respectively. The distribution of the difference values was more concentrated around zero in the model with the attention than the model without attention, indicating the superiority of the channel-wise attention model. In addition, the mean differences between the actual and the estimation (blue line in Figure 12) were 0.19 and 0.29 for the models with and without the attention, respectively. This indicates the high accuracy of the attention model, which means that this model has little bias compared to the model without attention.

#### 4.1.3. Optimal Sensor Analysis

The additional analysis was performed to determine the optimal sensors for the EE estimation based on the results presented above. First, one-way ANOVA was performed to investigate whether there is a significant difference among the average attention weights for each sensor calculated from the channel-wise attention. The results are shown in Table 3, where SS denotes the sum of squares, df denotes the degree of freedom, MS denotes the mean square, and F denotes the F-statistic. As a result of the ANOVA analysis, there was a statistically significant difference in the average attention weights of the sensors (p<0.001). Therefore, we also conducted a post hoc Tukey HSD test and the results are shown in Table 4. In the post hoc analysis, the symbols *P*, *A*, and *G* represent pressure, accelerometer, and gyroscope, respectively. The first letter in the subscript denotes the left (*L*) or right (*R*) side of the shoe, and the second letter is the detailed attachment position of the pressure sensors (see Figure 2) or the x, y, and z axis of the accelerometer and gyroscope. Each numerical value is an attention weight, and each column corresponds to a homogeneous subset with no statistically significant difference. For example, in column 1 of Table 4, there is no statistical difference in attention weights from $P_{L3}$ to $P_{L1}$ sensors (p>0.05), and the *p*-value of the corresponding subset is indicated at the bottom of the table. In Table 4, seven sensors are included in each subset from columns 1 to 6, but the number of sensors included in the subset sharply decrease in columns 7 to 10. This means that the sensors from columns 7 to 10 contributed significantly more to the EE estimation than those from columns 1 to 6. As a result, the subsets of sensors that are important for the EE estimation were $\left\{A_{LZ},G_{LZ},G_{LY}\right\}$, $\left\{G_{LY},A_{RZ}\right\}$, and $\left\{A_{RZ},G_{RZ}\right\}$ in the order of high attention weight. The accelerometer and gyroscope mostly show their higher contribution to the EE estimation than the pressure sensors, and particularly their attention weights in the z axis are higher than those in the other axes $G_{LY}$.

### 4.2. Heart Rate Estimation

#### 4.2.1. Proposed Model Performance

There are few previous studies conducted about the HR estimation using the pressure, accelerometer, or gyroscope sensors compared with those about the EE estimation because it is relatively easy to obtain an accurate heart rate using various off-the-shelf wearable devices equipped with physiological sensors such as electrocardiogram (ECG) and photoplethysmogram (PPG). However, users might be uncomfortable wearing an additional wrist or chest band to measure ECG or PPG. On the other hand, shoes could be a natural and unobtrusive wearable device to measure users’ physiological information. This study tried to extract the HR information from the pressure, accelerometer, and gyroscope sensors that were mounted on shoes due to the limitation of ECG and PPG measurements with high SNR by selecting the optimal sensors for the estimation. The performance of the heart rate estimation using the proposed model, which selects the optimal sensors with the help of the channel-wise attention mechanism, is shown in Table 5. Additionally, Figure 13 is the graph showing the actual and predicted values for the best and worst cases of the proposed model.

Accurately measuring the heart rate using physical sensors attached to smart shoes is challenging. Since the purpose of this study is to make it possible to easily measure heart rate in daily life, we compared it with the heart rate estimation accuracy of PPG-based wearable devices that are commercially available. Table 6 lists the accuracy of consumer wearable devices in heart rate estimation conducted by Nelson et al. [50]. The two devices that were compared were Apple Watch 3 (2017 version, Apple Inc, Cupertino, CA, USA, v. 4.2.3) and Fitbit Charge 2 (2017 version, Fitbit Inc, CA, USA, v. 22.55.2). In addition, MAE, Bland–Altman analysis, and mean absolute percent error (MAPE) were calculated as performance evaluation metrics. In particular, MAPE was calculated as follows:(15)$$MAPE=\frac{1}{N}\sum_{i=1}^{N}\left|\frac{y_{i}-{\hat{y}}_{i}}{y_{i}}\right|$$

In the previous study conducted by Nelson et al., the performance of each device under various conditions was evaluated. However, in Table 6, only the results in walking and running environments similar to this study were compared. The performance of the proposed model was 5.40 of MAPE, which is good compared with the results of Fitbit Charge 2 (9.21 and 9.88) and slightly worse than that of Apple Watch 3 (4.61 and 3.01).

#### 4.2.2. Channel-Wise Attention Effectiveness

The HR estimation with and without attention were also compared similar to EE estimation to verify the improvement of the performance using the channel-wise attention. The results are shown in Table 7. In HR estimation, the proposed model using the channel-wise attention achieved higher performance for all evaluation indicators including RMSE, MAE, and $R^{2}$, which are all statistically significant (p<0.05). This indicates that the channel-wise attention contributes to the selection of the optimal sensor for estimating the correct HR.

Figure 14 shows the Bland–Altman plot of HR estimation, where the LOA was in the range of [−15.12, 15.90] and [−21.46, 16.79] in the model with and without the attention, respectively, at the 95% confidence interval (±1.96SD). This indicates that the model with attention has less bias and higher stability than the model without attention, which is similar to the results of EE estimation.

#### 4.2.3. Optimal Sensor Analysis

One-way ANOVA was performed for HR estimation in the same way as EE estimation, for which the results are shown in Table 8. As a result of ANOVA analysis, there was a statistically significant difference in the averaged attention weight between the sensors (*p* < 0.001). In addition, a post hoc Tukey HSD test was conducted, and the results are shown in Table 9. In the post hoc analysis, the homogeneous subsets that contributed to the HR estimation were shown in the following order: $\left\{A_{LZ}\right\}$, $\left\{A_{RZ},G_{LY}\right\}$, $\left\{G_{LY},G_{LZ},G_{RZ}\right\}$. Same as the results of EE estimation, the accelerometer and gyroscope mostly showed a higher contribution than the pressure sensor and z axis direction sensors made a greater contribution than the other directions. In particular, the average attention weight of $A_{LZ}$ was significantly different from those of the other sensors, followed by $A_{RZ}$.

## 5. Discussion

In this study, it was shown that the proposed model could estimate the EE and HR using physical sensors such as accelerometer, gyroscope, and pressure sensors that can be equipped in smart shoes. In particular, the accuracy was improved with adaptively assigning weights to the sensors through the channel-wise attention, which is the core of the model to select the optimal sensors, making important contributions to the EE and HR estimations.

The proposed model shows that the z axis sensors in the accelerometer and gyroscope have higher contributions to the EE estimation than the others, as shown in Table 3 and Table 8. Among the previous EE estimation studies, Vathsangam et al. [23] calculated the EE in the treadmill while walking using an accelerometer sensor and a gyroscope sensor. They claimed that the x axis sensor in the accelerometer (y axis in this study) was aligned with the movement direction of the foot, indicating that its contribution to the EE estimation could be high. On the other hand, Javed et al. [51] found that the y and z axis features of the accelerometer were important to recognize walking and jogging activities. In another related study, Smith et al. [52] calculated the ratio of the triaxial to uniaxial (vertical) number in the accelerometer for various activities using an accelerometer sensor on the wrist. The results show that activities such as running are greatly affected by vertical movement. Moreover, we found that the average attention weight of the z axis was high corresponding to the running activity, which is largely affected by vertical activity. The findings of the significance of the z axis monitoring the vertical movement are consistent with the results of Javed et al. [51] and Smith et al. [52] since our study was conducted on a treadmill under similar conditions to the jogging activity.

In the HR estimation, the contributions of the z axis sensors in the accelerometer and gyroscope were high, which is similar to the results of EE estimation. In various previous EE estimation studies, the EE was directly calculated using the HR level [53]. However, in this study, the EE estimation was carried out separately from the HR estimation. As a result, large attention weights in the z axis in the proposed model seem to be significant considering the high correlation between HR and EE.

As an additional analysis, we performed ANOVA and post hoc analysis to verify whether there is a significant difference in attention weights among the x, y, and z axis sensors in the accelerometer and gyroscope. Figure 15 shows the average attention weight for each axis to predict the EE and HR levels. As a result, there was a significant difference between the x and z axes and between the y and z axes (p<0.001), although there was no statistical difference between the x and y axes.

## 6. Conclusions

In this study, the efficient HR and EE estimation models from multivariate raw signals including pressure, accelerometer, and gyroscope sensor data were designed using a deep learning architecture in an end-to-end manner. In addition, significant channels of the sensors were investigated using the channel-wise attention mechanism to estimate HR and EE, which found that the effects of the z axis sensors of the accelerometer and the gyroscope were significant in walking and running conditions. This is consistent with the previous study demonstrating that a general running activity is greatly affected by a vertical movement in the z axis direction [51,52]. This study also demonstrated the possibility of estimating HR and EE using the sensors mounted on shoes and suggests an effective and cost-efficient design of a wearable shoe-based device with selecting the optimal sensors. Furthermore, using the channel-wise attention, HR and EE were effectively estimated even when the individual left and right foot movements were not constant the during exercise. A limitation of this study is the small size of the training dataset and the individual characteristics of the participants with small deviations. Whilst the predictions might be a little unstable for datasets obtained under various conditions, the proposed model is trained and validated through the inter-subject analysis using LOSO, which could guarantee the generalizability of the proposed model if being adaptively retrained for each individual datum. Another limitation is that the computational load is large compared with the conventional approaches to estimate the HR and EE using a wrist band-typed photoplethysmogram (PPG) sensor (deep learning model size: approximately 70 mb, testing time: a few seconds). However, the existing HR and EE measurement devices have disadvantages when worn on a wrist, as some users feel uncomfortable to wear. In addition, they are too sensitive to noise, resulting in poor SNR. On the other hand, the proposed shoe sensor could be more natural for use to wear compared to the wrist-typed sensor.

For the future research, it would be possible to improve the generalization performance using more diverse datasets and adding personal information (gender, BMI, foot size, etc.) to the model input. It will also include the investigation of the sensor-specific functions corresponding to the variations in HR and EE values.

## Figures and Tables

**Figure 1 sensors-21-07058-f001:**
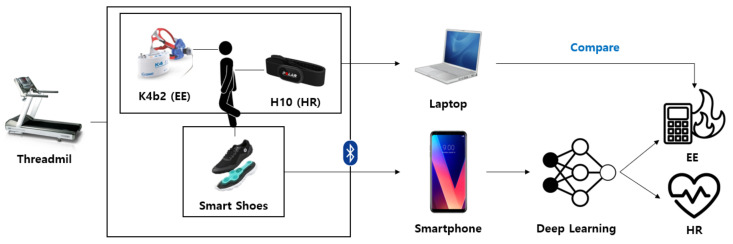
Overview of the system architecture for EE and HR estimation.

**Figure 2 sensors-21-07058-f002:**
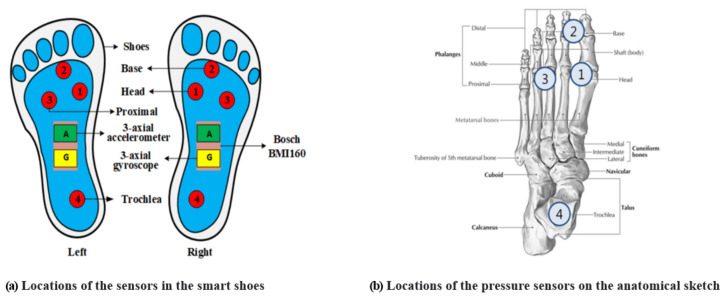
Locations of the sensors in the smart shoes: (**a**) a total of 12 sensors (6 sensors on the left and right shoe each) consisting of the pressure, accelerometer, and gyroscope sensors; (**b**) locations of the pressure sensors on the anatomical sketch: 1st metatarsal head (MH; sensor 1), toe (between the 1st and 2nd phalange; sensor 2), 4th metatarsal head (sensor 3), and heel (sensor 4).

**Figure 3 sensors-21-07058-f003:**
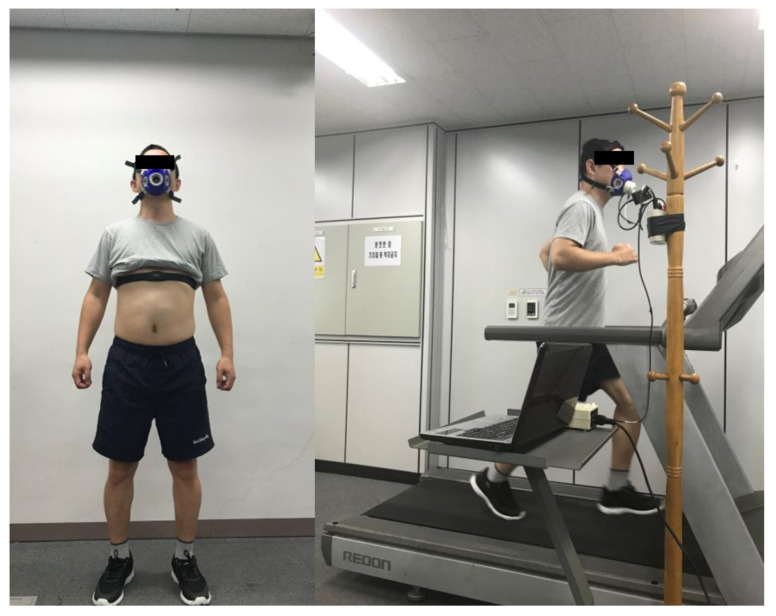
Example figures of the experimental equipment and process. During the experiment, participants wore a chest strap and a calorimeter to measure HR and EE, respectively. Each participant ran on a treadmill at a speed varying from 3 to 10 kph, which increased by 1 kph per every 2 min (total 16 min ran) and they were instructed to run at a constant speed as much as possible.

**Figure 4 sensors-21-07058-f004:**
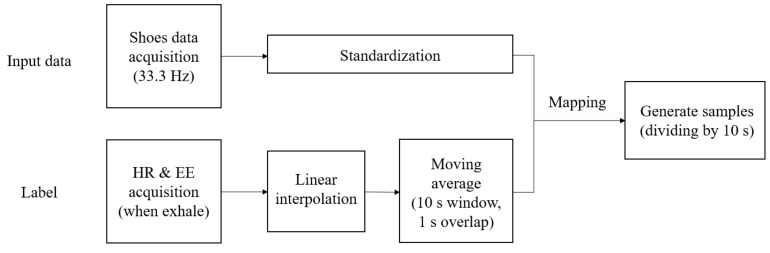
Flowchart of the data preprocessing when training the proposed deep learning model. The input shoes’ data were recorded at a 33.3 Hz sampling rate and standardized to have a zero mean and unit variance. The label was created based on the HR and EE information, which were averaged on a 10 s long window with an overlap of 1 s.

**Figure 5 sensors-21-07058-f005:**
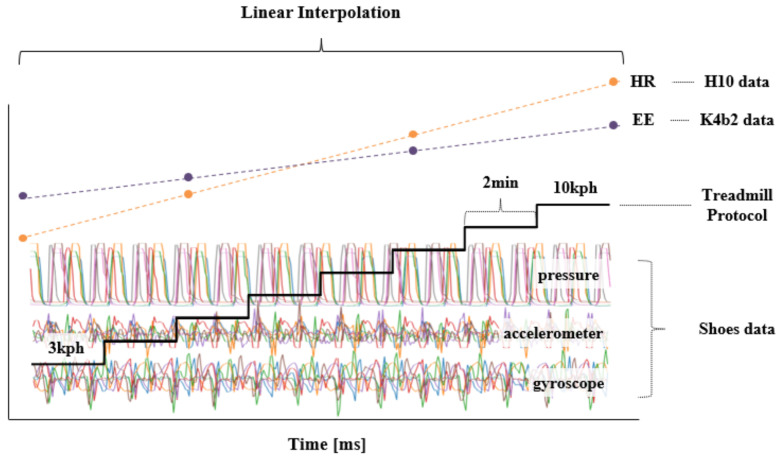
Application of a linear interpolation method due to the mismatch between the sampling rates of the HR/EE and data of the shoes’ sensors. In the HR and EE graphs, the green dot represents HR and the gold dot represents the EE of the actual measurement, and the dashed line is the estimated value.

**Figure 6 sensors-21-07058-f006:**
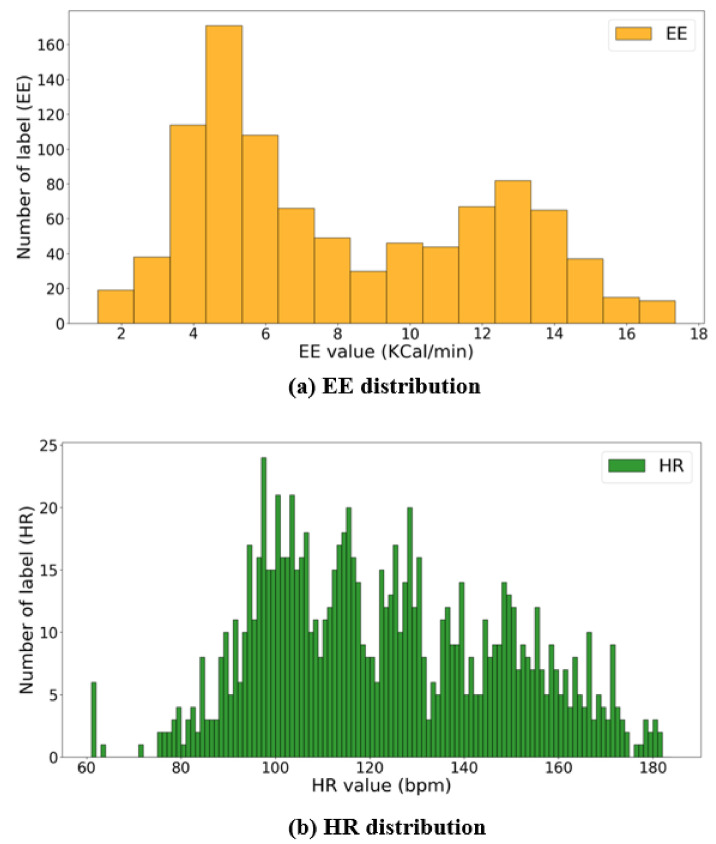
Distributions of HR and EE labels: (**a**,**b**) show the number of EE values per KCal/min and HR values per bpm, respectively.

**Figure 8 sensors-21-07058-f008:**
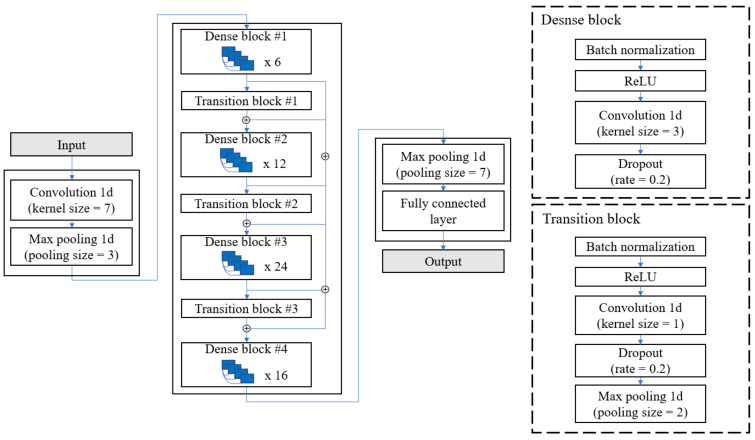
Internal structure of DenseNet. ⊕ denotes the concatenation of feature vectors.

**Figure 9 sensors-21-07058-f009:**
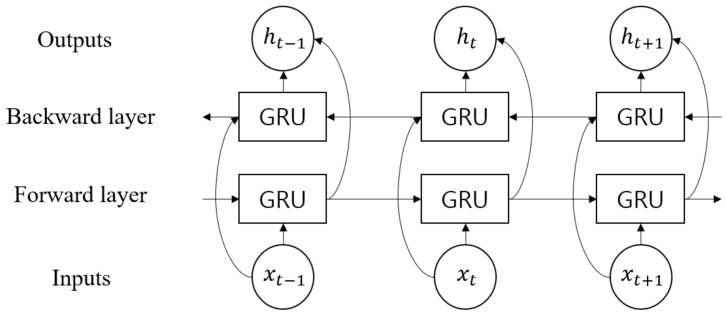
Structure of Bi-GRU.

**Figure 10 sensors-21-07058-f010:**
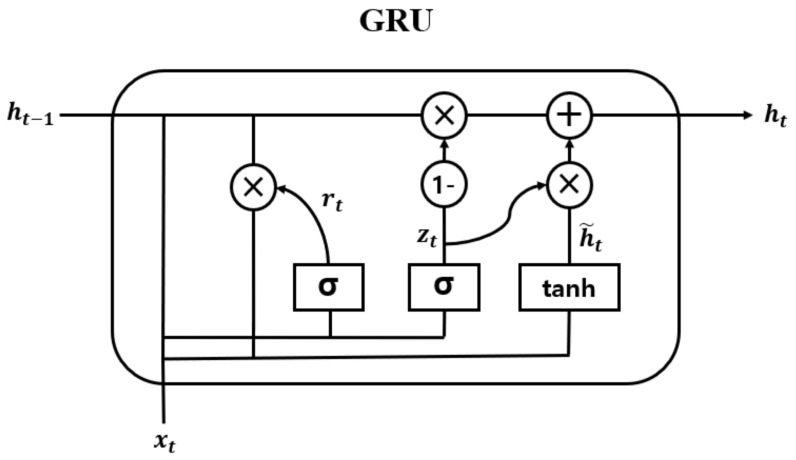
Internal structure of GRU.

**Figure 11 sensors-21-07058-f011:**
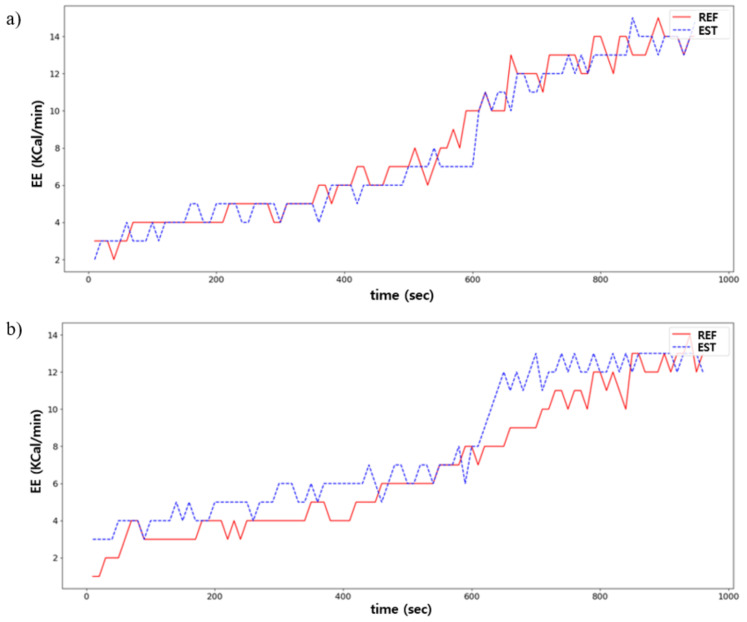
Comparison between the predicted (EST) and ground truths (REF) in EE estimation: (**a**) is the best case; (**b**) is the worst case.

**Figure 12 sensors-21-07058-f012:**
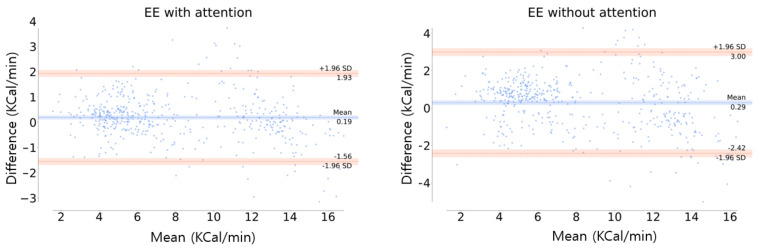
Bland-Altman plot of EE estimation. The orange line is the limit of agreement (LOA) and the center blue line is the mean of difference error between the actual and estimation.

**Figure 13 sensors-21-07058-f013:**
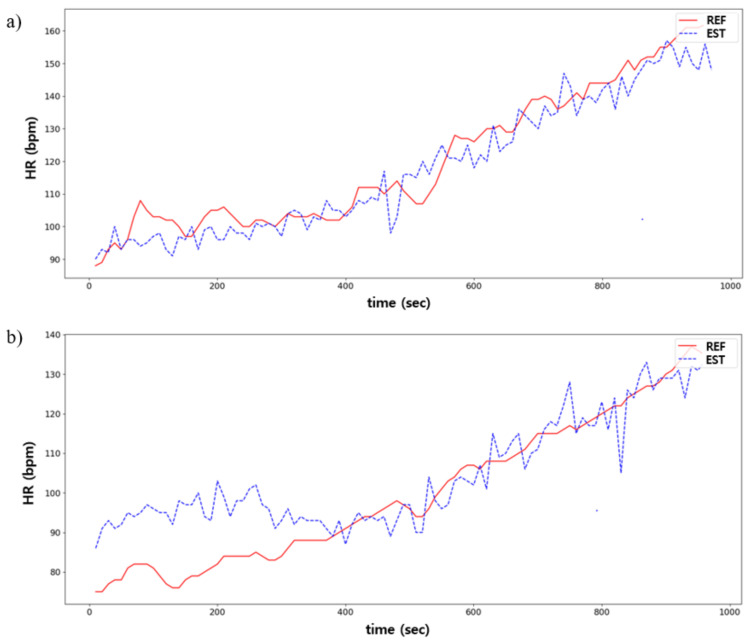
Comparison between predicted (EST) and ground truths (REF) in HR estimation: (**a**) is the best case; and (**b**) is the worst case.

**Figure 14 sensors-21-07058-f014:**
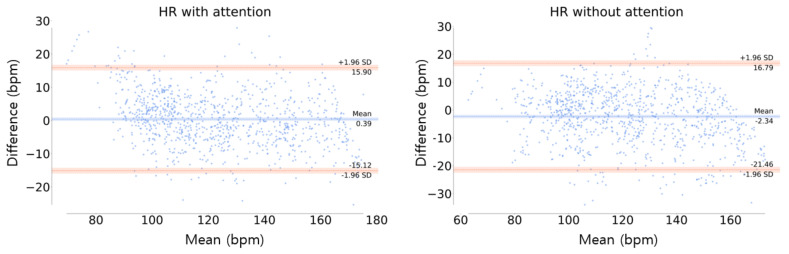
Bland–Altman plot of HR estimation. The orange line represents the limit of agreement (LOA) and the blue center line is the mean of the difference error between the ground truth and the estimation.

**Figure 15 sensors-21-07058-f015:**
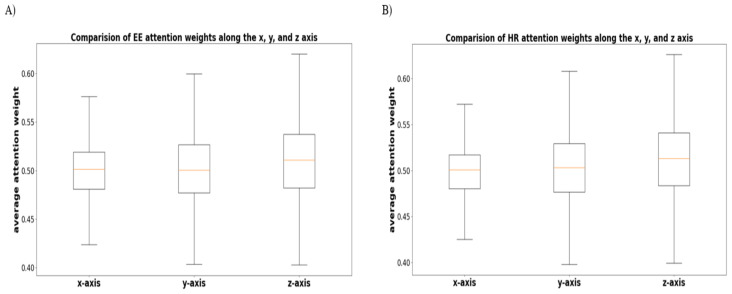
Comparison of the average attention weights for each of the x, y, and z axes. (**A**,**B**) illustrate the result of EE and HR, respectively.

**Table 1 sensors-21-07058-t001:** EE (KCal/min) estimation performance.

Input	RMSE	MAE	$R^{2}$
Acc + Gyro + Pr	1.05 ± 0.13	0.83 ± 0.12	0.922 ± 0.005

**Table 2 sensors-21-07058-t002:** Mean and standard deviation of RMSE, MAE, and $R^{2}$ values obtained using the proposed models with and without the attention mechanism in the EE estimation.

Input	RMSE	MAE	$R^{2}$
with attention (proposed)	1.05 ± 0.13	0.83 ± 0.12	0.922 ± 0.005
without attention	1.17 ± 0.24	0.95 ± 0.2	0.923 ± 0.12

**Table 3 sensors-21-07058-t003:** ANOVA analysis of the channel-wise attention weights in the EE estimation. SS is the sum of squares, df is the degree of freedom, MS is the mean square, and F is the F-statistic.

	SS	df	MS	F	*p*-Value
beetween groups	1.216	19	0.064	55.107	0.000
within groups	22.434	19,320	0.001		
total	23.649	19,339			

**Table 4 sensors-21-07058-t004:** Post-hoc Tukey HSD test result for the averaged attention weight for each sensor in EE estimation. Each column 1–10 represents a homogeneous subset for a significance level of 0.05. The sensor types are pressure (P), accelerometer (A), and gyroscope (G). The first subscript for each sensor type denotes the left (L) and right (R) sides of the shoe. The second subscript is the detailed attachment position of the pressure sensor (see Figure 2) or the x, y, and z axis directions of the accelerometer and gyroscope.

Sensor Type	1	2	3	4	5	6	7	8	9	10
$P_{L3}$	0.4908									
$P_{R2}$	0.4910	0.4910								
$P_{R4}$	0.4910	0.4910								
$P_{L4}$	0.4918	0.4918								
$G_{RY}$	0.4926	0.4926	0.4926							
$P_{R3}$	0.4935	0.4935	0.4935							
$P_{L1}$	0.4944	0.4944	0.4944	0.4944						
$A_{LY}$		0.4964	0.4964	0.4964	0.4964					
$P_{L2}$			0.4977	0.4977	0.4977	0.4977				
$A_{RX}$			0.4979	0.4979	0.4979	0.4979				
$A_{RY}$			0.4980	0.4980	0.4980	0.4980				
$P_{R1}$				0.4991	0.4991	0.4991				
$A_{LX}$				0.4998	0.4998	0.4998				
$G_{RX}$					0.4999	0.4999				
$G_{LX}$						0.5031	0.5031			
$G_{RZ}$							0.5070	0.5070		
$A_{RZ}$								0.5091	0.5091	
$G_{LY}$									0.5137	0.5137
$G_{LZ}$										0.5148
$A_{LZ}$										0.5155
*p*-value	0.742	0.069	0.057	0.060	0.750	0.072	0.546	0.999	0.240	1.000

**Table 5 sensors-21-07058-t005:** HR (bpm) estimation performance.

Input	RMSE	MAE	$R^{2}$
Acc + Gyro + Pr	7.81 ± 1.12	6.12 ± 0.86	0.897 ± 0.017

**Table 6 sensors-21-07058-t006:** Comparison of the HR estimation performance of commercial wearable devices and the proposed model. The performance of commercial wearable devices was cited from the study results of Nelson et al. [50].

Device	Condition	Device Error		Bland–Altman Analysis		
		MAE	MAPE	ME	Lower LOA	Upper LOA
Fitbit Charge 2	walking	9.55	9.21	−6.85	−28.51	14.81
	running	14.73	9.88	−14.73	−29.77	0.31
Apple Watch 3	walking	4.77	4.64	0.11	−14.18	14.41
	running	4.05	3.01	1.77	−9.78	13.33
proposed model	walking + running	6.12	5.40	0.39	−15.12	15.90

**Table 7 sensors-21-07058-t007:** Mean and standard deviation of RMSE, MAE, and $R^{2}$ values of models with (proposed model) and without the attention for HR estimation.

Input	RMSE	MAE	$R^{2}$
with attention	7.81 ± 1.12	6.12 ± 0.86	0.897 ± 0.017
without attention	9.19 ± 3.16	7.72 ± 3.67	0.878 ± 0.037

**Table 8 sensors-21-07058-t008:** ANOVA analysis of the channel-wise attention weights in the HR estimation. SS denotes the sum of squares, df denotes the degree of freedom, MS denotes the mean square, and F denotes the F-statistic.

	SS	df	MS	F	*p*-Value
beetween groups	2.145	19	0.113	90.706	0.000
within groups	24.049	19,320	0.001		
total	26.194	19,339			

**Table 9 sensors-21-07058-t009:** Post-hoc Tukey HSD test result for the average attention weight for each sensor in HR estimation. Each column 1–8 represents a homogeneous subset for a significance level of 0.05.

Sensor Type	1	2	3	4	5	6	7	8
$P_{R3}$	0.4864							
$P_{L3}$	0.4871							
$P_{R1}$	0.4888							
$P_{R4}$	0.4893							
$P_{R2}$	0.4900	0.4900						
$P_{L4}$	0.4901	0.4901						
$G_{RY}$	0.4911	0.4911	0.4911					
$A_{LX}$		0.4952	0.4952					
$A_{RY}$		0.4954	0.4954					
$P_{L1}$		0.4956	0.4956					
$A_{RX}$			0.4961					
$P_{L2}$			0.4963					
$G_{RX}$				0.5032				
$A_{LY}$				0.5064	0.5064			
$G_{LX}$				0.5067	0.5067			
$G_{RZ}$				0.5084	0.5084	0.5084		
$G_{LZ}$					0.5099	0.5099		
$G_{LY}$						0.5141	0.5141	
$A_{RZ}$							0.5167	
$A_{LZ}$								0.5229
*p*-value	0.288	0.061	0.122	0.130	0.794	0.056	0.986	1.000

## Data Availability

The data presented in this study are available on request from the corresponding author. The data are not publicly available due to ethical reasons.

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
