# Peer review of "Deep Learning-Based Optimal Smart Shoes Sensor Selection for Energy Expenditure and Heart Rate Estimation"

_sensors, 2021, doi:10.3390/s21217058_

Round 1

Reviewer 1 Report

The authors proposed an energy expenditure and heart rate estimation method based on deep learning architecture. The sensor attached to the shoes provides the channel data that can be feed into the model and gives a reasonable estimation of the EE and HR. Meanwhile, the channel-wise attention mechanism is used to identify the optimal sensor and optimize the weight to get a more accurate/stable estimation. The overall concept and method are well-defined. Some points can be considered before the acceptance of the manuscript.

  1. In the data preparation part, while the linear interpolation is more intuitive for the data resampling, does this method represent the actual scenario better compare to other interpolation method (e.g, polynomial)?
  2. With the relatively small training dataset and also small individual variation of the samples (similar age, height, weight, and foot size), it would be better to articulate the application range and robustness of the architecture with more details.
  3. One thing that appears to be interesting to explore is the data difference between the shoes. With both 20 channel data feed into the system, how does the estimation perform with the data from one shoe? If each 10 channel data has a large difference, how will this affect the estimation results? But it’s not a necessary point that needs to be addressed for this manuscript.

Reviewer 2 Report

In this study, the authors proposed an estimation of the energy consumption and heart rate in an environment  using smart shoes equipped with triaxial acceleration, triaxial gyroscope, and four-point pressure sensors.

Although the paper is quite well written, the results are clear but minor revisions should be made as follows:

Line 23-At the end of the paragraph, please insert the citation: "Ion, M.; Dinulescu, S.; Firtat, B.; Savin, M.; Ionescu, O.N.; Moldovan, C. Design and Fabrication of a New Wearable Pressure Sensor for Blood Pressure Monitoring. Sensors 202121, 2075. https://doi.org/10.3390/s21062075";

Line 98-Please specify the origin of the pressure, accelerometer, and gyroscope sensors. 

In section 2 Materials and Methods, the authors should present more data and pictures related to the measurement setup.

Insufficient information for Section 6 Conclusions. Please revise!

Add ANOVA in Abbreviations list.
